# Antioxidants and Therapeutic Targets in Ovarian Clear Cell Carcinoma

**DOI:** 10.3390/antiox10020187

**Published:** 2021-01-28

**Authors:** Tsukuru Amano, Atsushi Murakami, Takashi Murakami, Tokuhiro Chano

**Affiliations:** 1Department of Obstetrics & Gynecology, Shiga University of Medical Science, Tsukinowa-cho, Seta, Otsu, Shiga 520-2192, Japan; atsushim@belle.shiga-med.ac.jp (A.M.); tm@belle.shiga-med.ac.jp (T.M.); 2Department of Clinical Laboratory Medicine and Medical Genetics, Shiga University of Medical Science, Tsukinowa-cho, Seta, Otsu, Shiga 520-2192, Japan

**Keywords:** ovarian clear cell carcinoma, endometriosis, antioxidant, cancer stemness

## Abstract

Ovarian clear cell carcinomas (OCCCs) are resistant to conventional anti-cancer drugs; moreover, the prognoses of advanced or recurrent patients are extremely poor. OCCCs often arise from endometriosis associated with strong oxidative stress. Of note, the stress involved in OCCCs can be divided into the following two categories: (a) carcinogenesis from endometriosis to OCCC and (b) factors related to treatment after carcinogenesis. Antioxidants can reduce the risk of OCCC formation by quenching reactive oxygen species (ROS); however, the oxidant stress-tolerant properties assist in the survival of OCCC cells when the malignant transformation has already occurred. Moreover, the acquisition of oxidative stress resistance is also involved in the cancer stemness of OCCC. This review summarizes the recent advances in the process and prevention of carcinogenesis, the characteristic nature of tumors, and the treatment of post-refractory OCCCs, which are highly linked to oxidative stress. Although therapeutic approaches should still be improved against OCCCs, multi-combinatorial treatments including nucleic acid-based drugs directed to the transcriptional profile of each OCCC are expected to improve the outcomes of patients.

## 1. Introduction

Ovarian cancer is the eighth most common cancer affecting women worldwide, with an estimated 295,000 new cases in 2018 [1] and has a high mortality rate. In 2018 alone, approximately 185,000, 14,000, and 4800 deaths were reported worldwide, in the United States and Japan due to ovarian cancer, respectively [2,3]. Ovarian cancers can be classified into the following five pathological types: high-grade serous carcinomas (HGSCs), low-grade serous carcinomas (LGSCs), mucinous carcinomas (MCs), endometrioid carcinomas (ECs), and clear cell carcinomas (CCCs). The percentages of each pathological type among all ovarian cancers in the United States and Japan are as follows: serous carcinomas (combining HGSCs and LGSCs) 70%/36%, MCs 11%/11%, CCCs 5%/24%, and ECs 10%/17%, respectively. CCCs and ECs are much more common and great concerns in Japan and East Asian nations than in Europe or the United States [4,5]. HGSCs, which account for most of the serous carcinomas, are generally sensitive to chemotherapy. Furthermore, many studies have been conducted, providing new information. *TP53* mutations are detected in the majority of cases, and homologous recombination deficiency (HRD), including BRCA inactivation, have also been reported in approximately 50% of cases [6,7]. In this regard, novel treatments, such as the application of poly ADP-ribose polymerase (PARP) inhibitors, are being developed for HGSCs. LGSCs and MCs are relatively rare. LGSCs account for only 5–10% of ovarian serous carcinoma, and the majority of ovarian MCs are considered metastatic tumors derived from the gastrointestinal tract, while only ≤3% of MCs are regarded to be truly originating from the ovary [8,9]. 

Ovarian ECs and ovarian clear cell carcinomas (OCCCs) are both known as endometriosis-associated ovarian cancers, and are similar in that they are often associated with genetic mutations such as *ARID1A* and *PIC3CA*. Differences between these two are also apparent. ECs display estrogen and progesterone receptors [10] and are not significantly associated with antioxidant molecules such as hepatocyte nuclear factor 1 homeobox B (HNF1B) or mitochondrial superoxide dismutase (SOD2). The prognosis of ECs is relatively good, because they are sensitive to chemotherapy [11]. By contrast, OCCCs rarely express estrogen or progesterone receptors, and often overexpress HNF1B and SOD2. As overexpression of HNF1B and SOD2 confers resistance to oxidative stress in cancer cells, OCCCs possess strong resistance to oxidative stress caused by cancer treatments such as chemotherapy. Therefore, conventional anti-cancer drugs are ineffective against OCCCs, which often progress to advanced stages, and prognosis is extremely poor [12,13,14]. OCCC frequency is especially high in East Asian countries including Japan, and no effective treatment exists for OCCCs. As such, studies of molecular signatures of this type of cancer, and establishment of novel treatment methods, are urgently needed. 

Among the various pathological types of ovarian cancers, OCCCs are most strongly linked to oxidative stress tolerance. Oxidative stress causes cancer by directly damaging DNA. Recently, it has been elucidated that molecular abnormalities involved in oxidative stress generation are deeply involved not only in carcinogenesis but also in cancer progression, such as cancer invasion and metastasis. The involvement of oxidative stress in OCCCs can be divided into the following two categories: (a) carcinogenesis from endometriosis to OCCC and (b) factors related to treatment after carcinogenesis. 

This review focuses on the relationship between OCCCs and oxidative stress, the process and prevention of carcinogenesis, the nature of tumors, and treatment after carcinogenesis.

## 2. Linking Oxidative Stress and Carcinogenesis from Endometriosis to OCCCs 

A strong association between epithelial ovarian cancer and endometriosis, a common gynecologic disorder affecting approximately 10% of reproductive-age women [15], has long been suggested. Among epithelial ovarian cancers, OCCCs are considered the most closely associated with endometriosis. A pooled analysis of case-control studies indicated that a self-reported history of endometriosis is associated with a higher increased risk of OCCCs (odds ratio: 3.05) than the other histologic subtypes of ovarian cancers [16]. Additionally, the coexistent rate of endometriosis in the same ovary in each histologic subtype of ovarian cancer is reported to be 35.9% in OCCCs, 19% in ECs, 4.5% in SCs, and 1.4% in MCs [17]. 

Endometriosis contributes to the carcinogenesis of epithelial ovarian cancers through chronic inflammation, local hyperestrogenism, and oxidative stress. As represented by colitis-associated colorectal cancer, chronic inflammation has long been known to cause malignant transformation of cells and carcinogenesis [18,19]. Several inflammatory mediators, such as TNF-α, IL-6, and TGF-β, have been shown to participate in both the initiation and progression of endometriosis-associated ovarian cancers [20]. The abundance of estradiol in endometriotic lesions is caused by the locally increased expression of aromatase and steroidogenic acute regulatory protein combined with the decreased expression of 17β-hydroxysteroid dehydrogenase 2 [21]. These hormonal pathways are thought to be more closely associated with hormonal receptor-positive ovarian endometrioid carcinomas. On the other hand, as gene abnormalities associated with oxidative stress response and reactive oxygen species (ROS) metabolism are often detected [22], oxidative stress is considered to be most involved in the carcinogenesis of endometriosis into OCCCs. Endometriosis often results in chocolate (endometriotic) cyst formation, containing old blood with excess iron in the ovary. Iron and its metabolites contribute to the generation of ROS through the Fenton reaction, acting as inducers of DNA damage and sugar, lipid, and protein modifications, leading to carcinogenesis [23,24]. The carcinogenicity of iron compounds has been clearly demonstrated in previous animal experiments. Interestingly, several studies have shown that renal clear-cell carcinomas, which often share molecular features with OCCCs, were produced by intraperitoneal iron chelate injection [25,26]. Oxidative stress induced by excess heme production and iron accumulation could also be an important trigger in malignant transformation of endometriosis to OCCCs [27].

## 3. Attempts to Prevent OCCCs Developing from Endometriosis

In OCCCs, abnormalities are often found in genes associated with oxidative stress and ROS metabolism [22]. The elimination of persistent inflammation and ROS is important to prevent carcinogenesis from endometriosis to OCCC. Surgery is a useful tool to prevent endometriosis. A nested case-control study in Sweden revealed that compared to controls, a one-sided oophorectomy or radical extirpation of all visible endometriosis reduced the risk of later development of ovarian cancer to 19% and 30%, respectively. However, hormonal treatments, such as combined oral contraceptives, gestagens (including oral drugs or levonorgestrel-containing intrauterine devices), danazol, and gonadotropin-releasing hormone agonists, did not mitigate cancer risk [28]. Several cases of OCCCs arising from endometrioma during hormonal treatment have also been reported [29]. On the other hand, several studies reported that oral contraception reduces the risk of ovarian cancer among women with and without endometriosis by suppressing ovulation [30,31]. Through the collaborative analysis of data from 45 epidemiological studies, Basel et al. reported that after 5 years of oral contraceptive use, the risk of OCCC was reduced by 21.3% [31]. However, a few study limitations are attributed to the retrospective and contain data from areas with a low frequency of OCCCs among the population. Hormonal therapy using low-dose estrogen-progestin or dienogest (progestin medication) may suppress the progression of endometriosis to OCCC. A large-scale prospective study by the Japan Endometrioma Malignant-transformation Study (JEMS) is currently underway in Japan, where OCCCs are frequent in the population. Moreover, further research is needed to clarify this point.

Since oxidative stress is involved in the formation of OCCCs, antioxidant intake may be effective in preventing its development. VitaminA (carotenoid), C, E, flavonoids, and isothiocyanate are known antioxidant supplements [32,33,34,35]. Thus, diets rich in vegetables and fruits, which are good sources of antioxidants, are considered healthy. Antioxidants may prevent or delay various steps associated with carcinogenesis [36,37,38]. Carotenoid astaxanthin reduces oxidative stress, and inflammation [39,40] exerts a highly protective antioxidant effect [41]. Astaxanthin has been shown to decrease DNA damage and improve the immune response in healthy women after 8 weeks of intake [42]. In both in vitro and in vivo experiments, the use of astaxanthin significantly inhibited tumor formation and growth and exhibited anticancer properties [43,44,45,46]. Flavonoids inhibit multiple enzymes involved in cancer cell growth and arrest the cell cycle and tumor regression by activating the mitochondrial pathway of apoptosis [47,48]. Isothiocyanate inhibits the growth of ovarian cancer cells by inducing apoptosis in in vitro experiments [49]. Animal experiments have demonstrated that isothiocyanate exerts inhibitory effects on the carcinogenesis of both forestomach and lung cancers induced by the carcinogen benzopyrene [50]. 

As an epidemiological investigation, the effects of the daily intake of antioxidant supplements on ovarian cancer were investigated in multiple population-based case-control and prospective cohort studies, and several meta-analyses have been published. These studies indicated that the intake of dietary vitamins C, D, E, and isothiocyanate are not associated with the risk of ovarian cancers [51,52,53]. On the other hand, a higher dietary intake of vitamin A, including carotenoids and flavonoids, may lower the risk of ovarian cancer [54,55,56] (Table 1). In these studies, the subjects were members of the general population and did not necessarily have increased levels of oxidative stress. Antioxidant intake may be more effective in preventing carcinogenesis in people with endometriosis, as endometriosis treatment reduces the risk of developing ovarian cancers later. Because OCCC carcinogenesis is particularly affected by oxidative stress, antioxidants may be effective for its prevention. However, it should be noted that these studies did not consider the pathological type of ovarian cancer; thus, whether this leads to the prevention of OCCCs remains unknown. Furthermore, it must also be taken into account that the intake of vitamin A and beta-carotene has a negative effect on other carcinomas [54].

Currently, surgery is suitable to prevent carcinogenesis from chocolate cysts. However, the effects of hormonal therapies and dietary additives consisting of strong antioxidants, flavonoids, and isothiocyanates will require further investigation.

## 4. Molecular Characteristics in OCCCs Related to Anti-Oxidative Pathway

As mentioned above, in OCCCs, which arise from endometriosis under massive oxidative stress, abnormalities are often detected in genes associated with the oxidative stress response and ROS metabolism [22]. Several antioxidant molecules are involved in OCCC carcinogenesis. Among them, the overexpression of hepatocyte nuclear factor 1 homeobox B (HNF1B), a major homeobox-containing protein, also known as transcription factor-2, is highly important. Under hypoxia and acidosis, HNF1B can modify and adapt cancer cells to survive through a process between gluconeogenesis and glycolysis, commonly known as the Warburg effect [58]. Tsuchiya et al. [14] first reported HNF1B overexpression in OCCCs and showed that reduced HNF1B expression considerably increased the apoptosis rate in two OCCC cell lines. Overexpression of HNF1B was observed in endometrial tissues adjacent to OCCC tumors, suggesting that HNF1B overexpression is an early event in OCCC carcinogenesis. Kato et al. [59] found that hypomethylation of the CpG island of HNF1B induced its overexpression in OCCCs, indicating that overexpression in OCCC was also caused by epigenetic changes rather than by mutations. Moreover, recent research has revealed that HNF1B promotes the dedifferentiation of cancer stem-like cells (CSCs) via activation of the Notch pathway and enhancing the invasive potential and epithelial–mesenchymal transition in cancer cells [60]. Anti-oxidative pathways are deeply involved in carcinogenesis and therapeutic resistance in OCCCs. As oxidative stress tolerance represents therapeutic resistance, OCCCs usually exhibit poor and fatal prognoses, even during gradual progression. OCCC has low sensitivity to platinum- and taxane-based chemotherapy. Therefore, the prognosis of OCCCs is extremely poor, particularly in its advanced stages [61,62]. Previous studies have revealed the role of HNF1B in driving the expression of several characteristic genes associated with OCCCs [63], stimulating metabolic changes to promote gluconeogenesis, glycogen accumulation, and aerobic glycolysis [64], inducing chemotherapeutic resistance by suppressing sulfatase-1 (Sulf-1), an extracellular sulfatase catalyzing the 6-O desulfation of heparan sulfate glycosaminoglycans [65], and reducing the activity of immunological checkpoints against tumors. Thus, HNF1B plays an important role in therapeutic resistance via oxidative stress tolerance in OCCCs (Figure 1). 

Mitochondrial superoxide dismutase (SOD2) is an antioxidant enzyme that metabolizes superoxide in mitochondria and plays an important role in maintaining mitochondrial function through oxidative stress tolerance. SOD2 is highly expressed in the ectopic endometrium compared to normal endometrium, promoting cell proliferation and migration in ovarian endometriosis [66]. SOD2 is also highly expressed in OCCCs, and its oxidative stress tolerance appears to contribute to carcinogenesis [67,68]. SOD2 overexpression also promotes tumor growth and metastasis in OCCCs. Hemachandra et al. [67] found that SOD2 was more highly expressed in OCCCs than in any other epithelial ovarian cancer subtypes, and its overexpression contributes to tumor growth and metastasis in a chorioallantoic membrane model. The study also indicated that SOD2 expression was associated with increased cell proliferation, migration, outgrowth on collagen, spheroid attachment, and Akt phosphorylation in ES-2 OCCC cells. Therefore, SOD2 is regarded as a pro-tumorigenic or metastatic factor in OCCCs. Clinical studies have also demonstrated that high SOD2 expression was observed in 76% (33 out of 41) of OCCCs, and SOD2 overexpression was correlated with poor prognoses for OCCCs [68]. Accordingly, SOD2 is considered to be involved in therapeutic refraction through oxidative stress resistance in OCCCs (Figure 1).

## 5. Oxidative Stress and Cancer Stemness of OCCC

Recently, CSCs resistant to oxidative stress have been associated with the recurrence and metastasis of malignant tumors [69]. In other words, CSCs can survive severe oxidative stress induced by radiation therapy or chemotherapy and contribute to recurrence and metastasis. Because OCCC is often refractory to chemotherapy or relapse even after remission, it is suggested that CSCs are involved in the recurrence or metastasis of OCCC. High expression levels of aldehyde dehydrogenase 1 (ALDH1), a CSC marker, and Nrf2, a key transcriptional factor of the antioxidant system, were both associated with a poor prognosis in OCCC [70]. Furthermore, our group recently found that retinol dehydrogenase 10 (RHD10), enzymes related to vitamin A metabolism and gluconeogenesis, can reflect cancer stemness through precise analyses of the RAB39A (a member of the RAS oncogene family)‒RXRB (retinoid X receptor beta) axis. The RAB39A‒RXRB axis drives cancer stemness and tumorigenesis; consequently, the downregulation of this pathway leads to poor sphere formation and xenotransplantable function in several types of malignancies, such as sarcomas, adrenal, lymphoid, and testicular tumors [71]. On the other hand, some subpopulations of cancer cells could produce vividly growing spheres regardless of *RAB39A* repression. Therefore, under continuous *RAB39A* repression, we compared vividly growing cancer spheres to poor ones via RNA-seq transcriptional analysis, whose original sequencing data have been deposited in DDBJ/EMBL/GenBank (accession no. DRA010748). As a result, vividly growing cancer spheres were found to be significantly related to the upregulation of 79 genes (Appendix A) and of signaling pathways on retinoic acid (RA), vitamin A, and carotenoid metabolism, which were listed in statistically high orders of ranking (Table 2). In the pathway contributing to nuclear RXRB function (WP716_83589), retinol dehydrogenase 10 (RDH10), which converts retinol to all-trans RA and is indispensable for RA synthesis as the predominant enzyme [72,73], was focused on as a member of the 79 genes. 

Interestingly, RDH10 is involved in insulin signaling and contributes to gluconeogenesis via conversion from retinal to all-trans RA [74]. Gluconeogenesis and carbon hydrate storage are characteristics of OCCC phenotypes. Irrespective of RAB39A downregulation, RDH10 overexpression can result in the continuous activation of nuclear RXRB function and connect to CSC and carbon hydrate storage characteristics. OCCC cell lines specifically express abundant RDH10, rather than other types of ovarian cancer cells (Figure 2). RA is known to suppress cancer stemness and tumorigenesis because RA promotes cell differentiation, cell cycle arrest, and apoptosis via the heterodimer of retinoic acid receptor (RAR) and retinoid X receptor (RXR) [75,76,77]. In contrast, Schung et al. demonstrated that RA promotes cell survival in fatty acid-binding protein 5 (FABP5) cells through peroxisome proliferator-activated receptor beta (PPARβ/δ) [78]. Interestingly, the overexpression of FABP5 is an unfavorable prognostic marker in renal clear-cell carcinoma, which shows pathological similarities to OCCC [79]. Furthermore, RDH10 overexpression promotes tumor cell proliferation and correlates with patient survival time in gliomas [80]. 

Overall, RDH10 overexpression may imply cancer stemness and tumorigenesis in OCCC, resulting in a difficult prognosis refractory to existing therapies. RDH10 may serve as a novel diagnostic and therapeutic target for OCCC.

## 6. Therapeutic Targets for OCCCs in the Present and Future

As mentioned above, conventional standard treatments are less effective for OCCCs because of their strong tolerance to oxidative stress. Thus, to overcome the therapeutic difficulties associated with ovarian cancers, especially for OCCCs, novel therapeutics for recurrent or refractory cases are urgently needed. At present, several molecular targets have been proposed for OCCCs, which are categorized into the following groups: pathways related to receptor tyrosine kinases (RTKs), AT-rich interactive domain 1A (ARID1A)-related chromatin remodeling factors, and molecules associated with immune checkpoints. Some clinical trials have already been completed or are currently conducting, and we have summarized them (Table 3).

RTK receptors are located on the cell surface and play an important role in regulating cell proliferation, differentiation, survival, metabolism, and migration. Both the phosphoinositide 3-kinase/AKT/mammalian target of rapamycin (PI3K/AKT/mTOR) and epidermal growth factor/Ras/mitogen-activated protein kinase (EGF/Ras/MAPK) pathways are downstream pathways of RTKs. Mutations in phosphatidylinositol-4,5-bisphosphate 3-kinase catalytic subunit alpha (*PIK3CA*), the loss of phosphatase and tensin homolog (*PTEN*), the amplification of human epidermal growth factor receptor 2 (*HER2*), overexpression of *MET* (also known as hepatocyte growth factor receptor; *HGFR*), and ADP-ribosylation factor-like 4C (*ARL4C*) have been shown to activate these pathways in OCCCs [10,81,82,83]. In several studies, the inhibition of these molecules has shown the potential to suppress OCCCs. For example, MET inhibitors significantly decreased the proliferation and increased the apoptosis of OCCC cells in vitro, and suppressed tumor growth in xenograft models of OCCC in vivo [82]. Despite its effectiveness in vitro, no clinical advantages have been observed for inhibitors of downstream pathways of RTKs in treating OCCCs. The MET inhibitor cabozantinib was clinically ineffective in treating 13 patients with recurrent OCCCs [84]. The combination of temsirolimus and carboplatin or paclitaxel was also investigated in patients with advanced OCCCs. However, compared to conventional treatments, this regimen did not significantly increase the rate of progression-free survival [85]. Sunitinib, another RTK targeting inhibitor of VEGF and PDGF signaling, demonstrated minimal activity in second- and third-line treatments of persistent or recurrent OCCCs [86]. Thus far, no RTK inhibitors have demonstrated efficacy against OCCC in clinical trials. Further studies are needed to identify more effective drugs combined with PI3K/AKT/mTOR inhibitors and the mutations associated with OCCC that can be targeted by PI3K/AKT/mTOR inhibitors.

**Table 3 antioxidants-10-00187-t003:** Molecular targeting drugs and the clinical trials to treat OCCCs (ovarian clear cell carcinomas).

Category	Target Molecules	Clinical Research	Result	Reference
RTKs and related molecules	MET	already completed	minimal activity	[84]
	PI3K/AKT/mTOR	already completed	clinically ineffective	[85]
	VEGFR/PDGFR	already completed	minimal activity	[86]
	VEGFR/PDGFR/FGFR	Currently conducting	―	NCT02866370
ARID1A chromatin remodeling factor	EZH/glutathione	―	―	[87,88,89]
Immune checkpoint proteins	PD-L1	Currently conducting	―	NCT03405454
	PD-1/CTLA-4	Currently conducting	―	NCT03355976
	TIM-1	Currently conducting	―	NCT02837991

Clinical trials have been completed for several inhibitors of RTK-related pathways, but no significant effect against OCCCs has been found. Multiple clinical trials are currently underway to evaluate immune checkpoint inhibitors in OCCC.

ARID1A chromatin remodeling abnormalities are also useful therapeutic targets for OCCC [13]. The *ARID1A* gene encodes BAF250a, a subunit of the switch/sucrose non-fermentable (SWI/SNF) chromatin remodeling complex modifies the structure of chromatin via histone octamer ejection, octamer sliding, or local chromatin unwrapping to allow for the binding of other transcription factors [13]. Additionally, mutations in *ARID1A* contribute to AKT phosphorylation and induce PI3K pathway activation [90]. In *ARID1A*-mutated OCCCs, inhibition of the enhancer of zeste homolog 2 (EZH2) histone methyltransferase activity could induce synthetic lethality, including the suppression of PI3K/AKT signaling [87]. Additionally, Berns et al. [88] recently found that small-molecule inhibitors of the bromodomain and extra-terminal domain (BET) family of proteins inhibit the proliferation of ARID1A-mutated OCCC cells by reducing the expression of multiple SWI/SNF members in vitro and in vivo. Furthermore, recent research has demonstrated that *ARID1A*-deficient cancer cells have low levels of glutathione due to the decreased expression of SLC7A11 and are specifically vulnerable to the inhibition of the antioxidant glutathione and glutamate-cysteine ligase synthetase catalytic subunit, a rate-limiting enzyme for glutathione synthesis [89]. APR-246, a glutathione inhibitor, can act as an effective agent to induce synthetic lethality in *ARID1A*-deficient cancer cells [89]. Thus, EZH2, BET, and APR-246 are promising new drugs for the treatment of *ARID1A*-mutated OCCCs.

In recent years, significant progress has been made in targeting therapy to immune checkpoint proteins in cancers. In particular, immune checkpoint inhibitors (ICIs) by anti-programmed death receptor-1/programmed death-ligand 1 (PD-1/PD-L1) antibodies for high microsatellite instability (MSI-H) and high tumor mutation burden (TMB-H) tumors are effective in multiple studies [91,92,93,94]. The objective response rate and progression-free survival rate by PD-1 blockade were reported to be 40% and 78% for mismatch repair-deficient colorectal cancers and 0% and 11% for mismatch repair-proficient colorectal cancers, respectively [92]. Moreover, immunotherapy with PD-1 blockade for patients with advanced mismatch repair-deficient cancers across 12 different types achieved a 53% objective response rate and 21% complete response rate [91]. Furthermore, recent studies suggest that TMB can also predict the response to PD-1 inhibitors [93] and is associated with improved survival in patients receiving ICI across various cancer types [94].

As for OCCCs, recent research demonstrated that about 7% (4 out of 57) of OCCC cases had MSI-H cancers without any MMR mutations [95]. Feinberg reported that despite the presence of only four (1.6%) MSI-H tumors in 254 OCCC cases, 23 (9.0%) tumors with high TMB were found [96]. Additionally, it was also revealed that *ARID1A* alterations were associated with high TMB levels across cancer types and may cooperate with ICI treatment [97]. However, the relationship between *ARID1A*-mutated OCCC and efficacy of ICI treatment remains unknown. In summary, approximately 10% of OCCCs carry properties of MSI-H or TMB-H and may benefit from ICIs. For some OCCC patients, immunotherapy with anti-PD-1/PD-L1 antibodies has a high potential to be used as an effective treatment strategy. Multiple clinical trials are currently underway to evaluate PD-1/PD-L1 inhibitors in OCCCs. In tumor immunotherapy, another therapeutic candidate is T-cell immunoglobulin and mucin domain protein 1 (TIM-1), which regulates immune responses on human T cell surfaces. TIM-1, expressed on a high percentage of OCCC cells, exhausts T cell immunity in cancer microenvironments [98]. Clinical trials of CDX-014, an anti-TIM-1 antibody covalently linked to the potent cytotoxin, monomethyl auristatin E (MMAE), are currently being conducted in OCCC patients. Further studies of targeted therapies for immune checkpoint proteins are eagerly awaited.

As mentioned under the headings “Molecular characteristics in OCCCs related to anti-oxidative pathway” and “Oxidative stress and cancer stemness of OCCC,” molecules conferring oxidative stress resistance, including HNF1B, SOD2, and RDH10, are deeply implicated in therapeutic refractivity. Together with SOD2, HNF1B and RDH10 are major potential targets for OCCC treatment. However, molecular inhibitors of the latter two molecules have not been identified, and difficulties have arisen in development of chemical agents that inhibit them. In cancer cell subpopulations, especially in CSCs, the mitochondrial respiratory chain relies heavily on bioenergetic and biosynthetic processes. Therefore, mitochondrial function can be a target for therapeutic strategies in several cancers [86,99,100,101,102,103]. Since SOD2 plays an important role in maintaining mitochondrial function through oxidative stress tolerance, drugs suppressing mitochondrial function should be effective in treating SOD2-abundant OCCC. Replacement therapy or drug repositioning using biguanides, agents for treating diabetes mellitus that inhibit complex 1 of the mitochondrial respiratory chain may target tumor cell mitochondria and thereby improve the therapeutic effect on OCCCs [104]. It has been confirmed that metformin, commonly used as a first-line biguanide treatment for type 2 diabetes, can selectively inhibit mitochondrial respiratory chain complex 1 in various cancer cell lines [105,106,107,108]. As metformin is not metabolized, but almost all are excreted by the kidney, the half-life of plasma levels is long. Therefore, metformin accumulates in organs, and tissue concentration is considered to be higher than that of plasma [107]. Furthermore, many studies have shown that metformin accumulates at high concentrations in the mitochondria [107,109]. In addition to mitochondrial inhibition, metformin also demonstrates an anti-tumor effect through several routes, including immune-mediated, mammalian target of rapamycin (m-TOR), and AMP-activated protein kinase (AMPK) [110]. The follow-up program post-surgical resection of OCCC may involve drug repositioning using metformin; however, this will need to be verified by clinical cohort studies in the future. In addition, Molina et al. recently discovered IACS-010759, a new clinical-grade small-molecule inhibitor of complex I of the mitochondrial electron transport chain, which inhibits tumor growth in models of brain cancer and AML at well-tolerated doses in vivo [103]. SOD2-abundant OCCC may also be an adequate candidate for this new inhibitor. 

Targeting DNA helicases is another therapeutic strategy for MSI-H tumors, which may be suitable for use in combination with immune checkpoint inhibitors for cancer chemotherapy. Human DNA helicases RECQL1 and WRN proteins have been reported as therapeutic targets for several cancers, including ovarian cancers [111,112]. As for OCCCs, 9 out of 21 (43%) clinical cases showed high levels of RECQL1 expression by immunohistopathology, and RECQL1-siRNA significantly inhibited the proliferation of OCCC cell lines [112]. Importantly, recent research revealed that WRN induced double-stranded DNA breaks and promoted apoptosis and cell cycle arrest selectively in MSI-H cancer models [113]. In other words, WRN was selectively essential for MSI-H cancers, and the inhibition of WRN-induced synthetic lethality in MSI-H cancer models. These findings indicate that at least some OCCC patients benefit from RECQL1 or WRN inhibition.

In the future, further therapeutic options will need to be developed to improve OCCC achievements. In addition to drug repositioning against SOD2 anti-oxidative stress molecules and targeting RECQL1 and WRN to induce cancer-specific synthetic lethality, it is advisable to implement nucleic acid-based drugs, such as siRNA and antisense oligonucleotides, to treat OCCCs effectively and to adapt them to the transcriptional profile of individual tumor characteristics in the era of precision medicine. Integrative in silico approaches have indicated that only 10–15% of human proteins are druggable and that only 10–15% of human proteins are disease-modifying [114]. Hence, developing chemical agents that inhibit RDH10, HNF1B, and ARID1A is a challenge. However, considering the implementation of nucleic acid-based drugs, RDH10, HNF1B, ARID1A, RECQ, and WRN helicases should be included in future therapeutic targets. 

## 7. Conclusions

In this review, we have briefly summarized recent advances in the process and prevention of carcinogenesis, the characteristic nature of tumors, and the treatment of post-refractory OCCCs, which are highly linked to oxidative stress. The removal of oxidative stress suppresses the development of OCCCs in endometriosis. Strong antioxidants, such as vitamin A, carotenoids, or flavonoids, may help prevent carcinogenesis of OCCCs. However, the stress tolerance properties of OCCCs induce therapeutic resistance, making their treatment difficult. Antioxidants display bidirectional effects toward endometriosis and OCCCs. Elimination of oxidative stress, including by uptake of antioxidants, is highly effective in preventing progression from endometriosis to OCCCs, but, antioxidants are not suitable for treatment of established OCCCs, in which oxidative stress tolerance has accrued, providing therapeutic resistance. In OCCC therapeutics, inhibition of oxidative stress tolerance molecules is essential. The genetic and biological characteristics of OCCCs are being gradually evaluated, and the therapeutic effects of various anti-cancer drugs, molecular targeting drugs, drug repositioning strategies, and immunotherapies are being verified. Further studies will be needed to identify novel molecular targets, and studies on precision medicine, combining multiple treatments based on the genetic and molecular characteristics of individual tumors, will need to be conducted. Since the development of small molecular inhibitors for some undruggable molecules remains a challenge, it is essential that therapeutic approaches against OCCCs be improved, and that nucleic acid-based drugs and multi-combinatorial treatments corresponding to the transcriptional profile of each tumor be implemented.

## Figures and Tables

**Figure 1 antioxidants-10-00187-f001:**
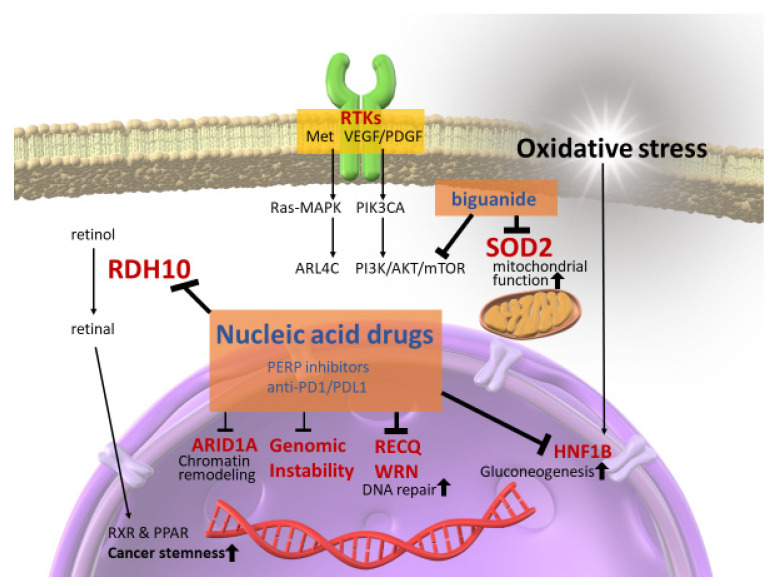
Activating pathways and targeting proposals in ovarian clear cell carcinomas. In ovarian clear cell carcinomas, downstream of receptor tyrosine kinases (RTKs), AT-rich interactive domain 1A (*ARID1A*)-related chromatin remodeling factors, and genomic instability, including MSI-H, are activated. These are currently being targeted. However, other therapeutic strategies, such as nucleic acid-based drugs, RDH10, RECQL1, WRN, and HNF1B, should be targeted in the future to reduce cancer stemness, induce cancer-specific synthetic lethality, and reduce gluconeogenesis, together with a drug repositioning strategy against SOD2 anti-oxidative stress molecules.

**Figure 2 antioxidants-10-00187-f002:**
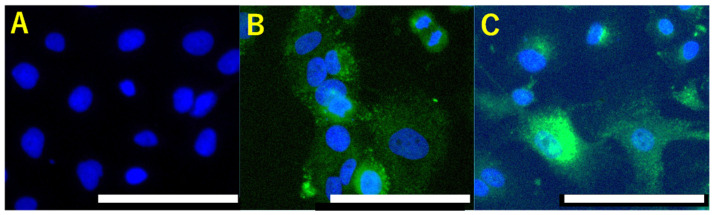
Abundant retinol dehydrogenase 10 (RDH10) expression in ovarian clear cell carcinomas. In contrast to SK-OV-3, ovarian serous cell carcinoma (**A**), ovarian clear cell carcinoma cells, OVISE (**B**), and TOV-21 (**C**) express high levels of retinol dehydrogenase 10 (RDH10). Scale bars: 100 μm.

**Table 1 antioxidants-10-00187-t001:** Antioxidant supplements and foods against ovarian cancers.

Clinical Research	Antioxidant	Result	Reference
Meta-analysis	Vitamin A	Reduced the risk of ovarian cancer.	[52]
Meta-analysis	Vitamin C	No significant effect on the risk of ovarian cancer.	[53]
Systematic review	Vitamin D	No significant effect on the risk of ovarian cancer.	[55]
Meta-analysis	Vitamin E	No significant effect on the risk of ovarian cancer.	[51]
Meta-analysis	Flavonoid	Reduced the risk of ovarian cancer.	[56]
Cohort study	Flavonoid Isothiocyanate	Reduced the risk of ovarian cancer. No significant effect on the risk of ovarian cancer.	[57]

A recent systematic review, cohort study, and meta-analyses indicated that vitamin A and flavonoid intake might reduce the risk of ovarian cancers, although vitamins C, D, E, and isothiocyanate have no significant effects on the risks. The preventive effects of flavonoids and vitamin A on OCCC (ovarian clear cell carcinoma) remain unclarified.

**Table 2 antioxidants-10-00187-t002:** Upregulated pathways in vividly growing cancer spheres.

Pathway	*p*-Value (RNAseq from Vividly Growing Cancer Spheres)
Hs_Integrated_Cancer_Pathway_WP1971_82939	1.02 × 10^4^
Hs_Intrinsic_Pathway_for_Apoptosis_WP1841_83332	0.002799811
Hs_Signaling_by_Retinoic_Acid_WP3323_83286	0.003082763
Hs_Vitamin_A_and_Carotenoid_Metabolism_WP716_83589	0.003229062
Hs_BMAL1-CLOCK, NPAS2_activates_circadian_gene_expression_WP3355_83343	0.003687094
Hs_Integrated_Breast_Cancer_Pathway_WP1984_82941	0.004416806
Hs_Pre-NOTCH_Expression_and_Processing_WP2786_83418	0.004513508
Hs_Lipid_storage_and_perilipins_in_skeletal_muscle_WP2887_85092	0.011662396
Hs_Uptake_and_function_of_anthrax_toxins_WP3390_83389	0.013593029
Hs_miRNA_Regulation_of_DNA_Damage_Response_WP1530_84694	0.013785134

Pathways were explicitly upregulated in vividly growing cancer spheres, irrespective of continuous RAB39A repression. Note: The pathway of retinoic acid signaling, namely vitamin A and carotenoid metabolism, is listed in the superior order of the top 10 rankings.

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
