# Peer review of "Antioxidants and Therapeutic Targets in Ovarian Clear Cell Carcinoma"

_antioxidants, 2021, doi:10.3390/antiox10020187_

Round 1

Reviewer 1 Report

This review describes the carcinogenesis, characteristic nature, and therapeutic strategy of ovarian clear cell carcinomas (OCCCs) from a molecular point of view relating with oxidative stress. This review is comprehensive and helps readers to understand the molecular mechanism in OCCCs. This article is in good quality.

Author Response

Thank you very much for your appreciation of our manuscript. We were pleased to learn that our manuscript was rated as acceptable for publication.

The comments raised by the other reviewers were given attention, and we have revised our manuscript, indicating the corrections with red letters. Please refer them. In addition, we shall give our permission for any language corrections you intend to make.

We hope you will find the revised version to be continuously acceptable for publication in “Antioxidants Special Issue: Antioxidants and Cancer".

Reviewer 2 Report

This review summarizes the potential connection between oxidative stress and endometriosis associated ovarian clear cell carcinoma. This review also provides suggestions of using antioxidants to improve outcomes in these patients.

Though this is a well-written review, there are several areas that needs improvement.

  1. The table 1 shows overexpression of antioxidant genes in OCCC. Hence it should be clarified if oxidative stress should be inhibited or activated in OCCC.
  2. It has to be made very clear to the readers if oxidative stress is involved in endometriosis leading to OCCC. If so, blocking oxidative stress in patients with endometriosis not OCCC will prevent endometriosis leading to OCCC.
  3. If this is so, an increase in antioxidant genes as shown in Table 1 would suggest possible interference with treatment. This again has to be clarified in a better way. Is oxidative stress low in patients with OCCC thus leading to higher antioxidant defense. Is this increase in antioxidant defense leading to loss of response to treatment? If so, will blocking of the antioxidants, or increase in oxidative stress assist in better outcomes in OCCC.
  4. If this is not clarified, then it is difficult to conclude if oxidative stress is good or bad in OCCC and if antioxidants should be used in patients with OCCC.

Author Response

Thank you very much for your constructive comments. The comments raised by you were given attention and the manuscript was revised.

1. The table 1 shows overexpression of antioxidant genes in OCCC. Hence it should be clarified if oxidative stress should be inhibited or activated in OCCC.

We have fully understood your curiosity. In general, oxidative stresses caused by traditionally anti-cancer agents inhibits cancer cells. However, overexpression of antioxidant molecules such as HNF-1B and SOD2 confers the resistance to oxidative stresses and chemotherapeutic resistance in OCCCs.

As we have deleted “previous Table 1” according to the comment of Reviewer #3, the descriptive sentence has been added in the introduction section (Page 2, Line #56-60).

2.  It has to be made very clear to the readers if oxidative stress is involved in endometriosis leading to OCCC. If so, blocking oxidative stress in patients with endometriosis not OCCC will prevent endometriosis leading to OCCC.

We have appreciated to your curiosity. As you suggested, we have clearly described for the readers that oxidative stress is involved in endometriosis leading to OCCCs in the last sentence of “2. Linking oxidative stress and carcinogenesis from endometriosis to OCCCs” section (Page 3, Line #103-105).

3-4.  If this is so, an increase in antioxidant genes as shown in Table 1 would suggest possible interference with treatment. This again has to be clarified in a better way. Is oxidative stress low in patients with OCCC thus leading to higher antioxidant defense. Is this increase in antioxidant defense leading to loss of response to treatment? If so, will blocking of the antioxidants, or increase in oxidative stress assist in better outcomes in OCCC.

If this is not clarified, then it is difficult to conclude if oxidative stress is good or bad in OCCC and if antioxidants should be used in patients with OCCC.

Thank you again for your curiosities.

The effects of antioxidants on endometriosis and OCCCs are bidirectional. That is, elimination of oxidative stress, including intake of antioxidants, is highly effective to prevent the progression from endometriosis to OCCCs, while antioxidants are not suitable for the treatment of OCCCs, in which oxidative stress tolerance has been occurred and been causing therapeutic resistance. In OCCC therapeutics, the inhibition of oxidative stress tolerance molecules is essentially required.

Therefore, we have added the above-mentioned descriptions in the conclusion section (Page 10, Line #427-432).   

Your kind consideration of our manuscript would be greatly appreciated. We hope you will find this version acceptable for publication in “Antioxidants Special Issue "Antioxidants and Cancer".

Reviewer 3 Report

Reviewer to Antioxidants

This is a review study about ovarian clear cell carcinomas. The authors try to summarize the recent advances in the process and strategies to prevent carcinogenesis, the characteristic nature of tumors, and the treatment of post-refractory OCCCs, which are highly linked to oxidative stress.

There is a lot of content to be described in a 10 pages manuscript. Sometimes it is too much superficial, but they did a general narrative over the article.

Title: So broad. Need more focus and clear idea what the readers could expect to find.

Introduction:

Page 2, Line 2 said: “Various comparisons of these two parameters are shown in Table 1”. Please, be more specific.

Table 1 said: Comparison of genetic, biological, and clinical features of endometriosis-associated ovarian carcinomas. There are only few genetic features specified in the table. I suggest expanding this table or only describe the 2 lines in the main text.

Why there is an explanation/description after the title of the tables? Please correct it along the manuscript.

The authors mentioned that EC and OCCC are linked with endometrioses, they described the genetic features but then they decided to focus on OCCC without mention how is the oxidative stress in EC that they were describing before. Even tough they mentioned that OCCC is the most tumor associated with oxidative stress, how is EC? Please do a brief comment.

Page 3, line 14: The authors mentioned ROS. They will describe it as reactive oxygen species (ROS) on page 3 line 18. Please correct it and place the description in the first citation of the acronym.

Page 5: HNF1B is gene or protein expression? If gene, please put gene symbol in italic.

CSC was defined on page 5 line 14 and page 6 line 21. Please correct it.

Page 6, line 45: RDH family or RDH10? Please clarify.

Topic 6. Therapeutic targets for OCCCs in the present and future: I suggest a table with the current drugs and recent approval clinical trials or a figure showing the targets and molecules that have been used as strategies to treat OCCCs.

References:

For a review manuscript I expected recent literature specially for treatment approach. Is there some recent publication about it?

Author Response

Thank you very much for your constructive comments. The comments raised by you were given attention and the manuscript was revised.

1.  Title: So broad. Need more focus and clear idea what the readers could expect to find.

According to your advice, we modified the title as follow:

“Antioxidants and therapeutic targets in ovarian clear cell carcinoma”

2.  Introduction: Page 2, Line 2 said: “Various comparisons of these two parameters are shown in Table 1”. Please, be more specific. I suggest expanding this table or only describe the 2 lines in the main text.

Following to your constructive comment, we have deleted table1 and its contents have been included in the manuscript (Page 2, Line #48-62). We have to add an explanation focusing on OCCCs rather than ECs, and have been also demanded as a relatively long and general review by the Editorial official. Then, although the length of our revised manuscripts has relatively long, we would appreciate it if you would refer and consider the matter. 

3.  Why there is an explanation/description after the title of the tables? Please correct it along the manuscript.

Thank you for pointing that out. We have checked and corrected all the tables as your pointing out.

4.  The authors mentioned that EC and OCCC are linked with endometrioses, they described the genetic features but then they decided to focus on OCCC without mention how is the oxidative stress in EC that they were describing before. Even though they mentioned that OCCC is the most tumor associated with oxidative stress, how is EC? Please do a brief comment.

Following to your suggestion, we have added a brief comment about the oxidative stress in ECs into the introduction section (Page 2, Line #51-53).

5.  Page 3, line 14: The authors mentioned ROS. They will describe it as reactive oxygen species (ROS) on page 3 line 18. Please correct it and place the description in the first citation of the acronym.

According to your comment, we have made the corrections (Page 2, Line #94; Page 2, Line #98).

6.  Page 5: HNF1B is gene or protein expression? If gene, please put gene symbol in italic.

Thank you for your question. HNF1B is shown as a protein expression in these cases. Therefore, we did not change them.

7.  CSC was defined on page 5 line 14 and page 6 line 21. Please correct it.

According to your comment, we have made the correction (Page 6, Line #225).

8.  Page 6, line 45: RDH family or RDH10? Please clarify.

Thank you for your question. RDH10 is indispensable for RA synthesis as the predominant enzyme (PMID: 28207193, 27983671). The following additions have been made into the section “5. Oxidative stress and cancer stemness of OCCC” (Page 6, Line #248-249) and the references [#72, 73].

9.  Topic 6. Therapeutic targets for OCCCs in the present and future: I suggest a table with the current drugs and recent approval clinical trials or a figure showing the targets and molecules that have been used as strategies to treat OCCCs.

References: For a review manuscript I expected recent literature specially for treatment approach. Is there some recent publication about it?

Following to your suggestion, we have created a new table (Table 3; Page 8, Line #312-315) and added some descriptions into the section “6. Therapeutic targets for OCCCs in the present and future” (Page 7, Line #285-287; Page 8, Line #306-309; Page 8, Line #336-337; and Page 9, Line #3658-365) and the additional reference [#90].

Your kind consideration of our manuscript would be greatly appreciated. We hope you will find this version acceptable for publication in “Antioxidants Special Issue "Antioxidants and Cancer".

Reviewer 4 Report

The publication summarizes the progress in the process and the prevention of carcinogenicity, describes the characteristics of tumors and the treatment of post-resistant OCCC, which are strongly associated with oxidative stress. Currently, there is still a search for a new therapeutic approach that should be improved over OCCC, with the expectation that multu-compenent treatments, including nucleic acid-based drugs, will improce patient outcomes. In this review, the authors mainly highlighted the relationship between OCCCs and oxidative stress and carcinogenesis, the nature of cancer itself, and treatment.Good work demonstrating the need for further research and development of further therapeutic options to improve treatment outcomes for OCCCs. Further studies will be needed to identify novel molecular targets as well as studies on precision medicine.

Author Response

(The authors gave the same response as above.)

Round 2

Reviewer 2 Report

Thank you for revising your manuscript.